# Repurposing Mouthwashes: Antifungal and Antibiofilm Abilities of Commercially Available Mouthwashes Against *Candida* spp.

**DOI:** 10.3390/antibiotics14050483

**Published:** 2025-05-09

**Authors:** Marie Maziere, Paulo Rompante, José Carlos Andrade, Beatriz S. F. De Oliveira, Mariana C. Alves, Celia Fortuna Rodrigues

**Affiliations:** 1UNIPRO—Oral Pathology and Rehabilitation Research Unit, University Institute of Health Sciences (IUCS-CESPU), Avenida Central de Gandra 1317, 4585-116 Gandra PRD, Portugal; marie.maziere@iucs.cespu.pt (M.M.); paulo.rompante@iucs.cespu.pt (P.R.); 2Associate Laboratory i4HB—Institute for Health and Bioeconomy, University Institute of Health Sciences (IUCS-CESPU), Avenida Central de Gandra 1317, 4585-116 Gandra PRD, Portugal; jose.andrade@iucs.cespu.pt; 3UCIBIO—Applied Molecular Biosciences Unit, Forensics and Biomedical Sciences Research Laboratory, University Institute of Health Sciences (1H-TOXRUN, IUCS-CESPU), Avenida Central de Gandra 1317, 4585-116 Gandra PRD, Portugal; 4Microbiology Laboratory, University Institute of Health Sciences (IUCS-CESPU), Avenida Central de Gandra 1317, 4585-116 Gandra PRD, Portugal; a31323@alunos.cespu.pt (B.S.F.D.O.); a32672@alunos.cespu.pt (M.C.A.); 5ALiCE/LEPABE—Laboratory for Process Engineering, Environment, Biotechnology and Energy, Rua Dr. Roberto Frias, 4200-465 Porto, Portugal

**Keywords:** *Candida* spp., oral candidiasis, mouthwash, repurposable drugs, chlorhexidine digluconate, cetylpyridinium chloride

## Abstract

**Background/Objectives**: The main objective was to evaluate and compare the antifungal efficacy against *Candida* spp. in commercially available mouthwashes distributed in the European market. Indeed, the solution to emerging infectious diseases may no longer lie in costly new drug development but rather in unlocking the untapped potential of existing substances. **Materials and Methods**: Eighteen mouthwashes, chosen based on their composition, were tested in vitro against ten *Candida* strains, including clinical isolates of oral origin and reference strains, in both planktonic and biofilm forms. The antifungal susceptibility testing was conducted using the European Committee on Antimicrobial Susceptibility Testing (EUCAST) disc diffusion method and the evaluation of the kinetic growth in planktonic *Candida*. Biofilm reduction was determined by the evaluation of the minimal biofilm eradication concentration (MBEC). Scanning electron microscopy (SEM) analysis was performed to evaluate potential morphological alterations of *Candida* biofilms. **Results**: Most mouthwashes effectively reduced biomass production and colony-forming unit (CFU) count. Parodontax Extra showed the highest efficacy. In the disc diffusion assay, Gum Paroex 0.12% exhibited the largest average inhibition zone diameter. Some unusual trends in the data may be explained by a higher reaction of fungal cells and the release of excess biomass during co-incubation in higher concentration of mouthwashes. SEM images revealed significant morphological alterations. **Conclusion**: Mouthwashes containing chlorhexidine digluconate, either alone or in combination with cetylpyridinium chloride and other active compounds, emerged as a common factor among the most efficacious formulations. In vivo studies will be essential to validate these findings, but mouthwashes may serve as a valuable adjuvant in the treatment of oral candidiasis.

## 1. Introduction

What if the key to fighting emerging infectious diseases lies not in developing expensive new drugs but in reimagining the potential of substances already within our reach? The ‘repurposable drugs’ approach fits into a world that needs to save resources and find quick solutions: it is indeed the search for new therapeutical indications for existing drugs [1,2]. In fact, this approach has already been explored by some researchers. Recent studies have explored the antifungal properties of non-steroidal anti-inflammatory drugs (NSAIDs) like ibuprofen and diclofenac, which have shown inhibitory effects against *Candida* species and other fungal pathogens [3,4]. Additionally, anticancer drugs, such as tamoxifen and bleomycin, have demonstrated antifungal activity, suggesting that their repurposing could provide alternative treatment options for fungal infections [5,6].

Infectious diseases present additional challenges in developing new treatments due to the ability of microorganisms to acquire resistance through genetic mutations and after exposure to antimicrobial agents, making it a battle that never ends [7]. In fact, commonly used antifungal drugs in dentistry—such as nystatin, fluconazole, and miconazole—have increasingly been shown to be less effective in certain cases in recent years [8]. In response to this global health urgency, in 2022, the World Health Organization (WHO) released a landmark report highlighting a list of priority pathogenic fungi that require urgent attention to enhance the global response to fungal infections and antifungal resistance [9,10].

Among fungal infections, oral candidiasis appears among the most common worldwide. Oral candidiasis is an infection caused by fungi of the genus *Candida*, normally harmless commensals in the gastrointestinal and genitourinary tracts in about 70% of humans but may turn pathogenic in the oral cavity in dysbiosis conditions [11,12]. About 90% of *Candida* infections are caused by five species identified by the WHO’s report in 2022: *Candida albicans*, *Nakaseomyces glabrata*, *Candida parapsilosis*, *Candida tropicalis*, and *Pichia kudriavzevii* [13]. Some of these microorganisms are particularly strong because of their ability to form biofilms to enhance their survival, secreting an extracellular matrix to increase their adhesion to mucosa and gingiva, and increasing invasion by their ability to create hyphae and pseudo-hyphae, called polymorphism and/or enzyme production (e.g., proteases) [14,15].

Given the urgency highlighted by the WHO, the huge potential of repurposing drugs led to an investigation into the antifungal effects of common mouthwashes. These readily available products, already approved for human use, can be explored for their efficacy against oral fungal infections. Oral care products are used daily by billions of people all around the world [16,17]. The market is extensively supplied with various types of mouthwashes (also called mouth rinse or mouthrinse) for diverse functions. Among the products used to “treat gums and gingiva” are mouthwashes that can contain diverse antimicrobial agents either individually or in combination, such as chlorhexidine digluconate, cetylpyridinium chloride, hexetidine, sodium fluoride, menthol, eucalyptol, and thymol, among others [18,19].

The main objective of this study was to evaluate and compare the antifungal efficacy against *Candida* spp. in commercially available mouthwashes distributed in the European market. The specific research objectives encompass the selection of mouthwashes; a comprehensive chemical composition analysis of the selected mouthwashes; and the assessment and comparison of the antifungal properties exhibited by these mouthwashes against *Candida* spp., involving reference strains and isolates derived from the oral cavity of patients, both in planktonic cells and biofilm.

## 2. Results

### 2.1. Mouthwash Composition

The detailed composition of mouthwashes was analysed on the basis of the information provided by the manufacturers on the labels of the products we analysed (Table 1). The antiseptic components of each product were thoroughly analysed (Figure 1).

### 2.2. Antifungal Susceptibility

#### 2.2.1. Disc Diffusion Assay

After a 24 h incubation period, every mouthwash showed inhibition except for Alodont, Arthodont, and Listerine Cool Mint (Figure 2). Gum Paroex 0.12% exhibited the largest average inhibition zone diameter, followed by Bexident Post 0.20% and Eludril Classic (tied for second), with Periogard, PerioPlus Curaprox, and Parodontax 0.10% showing progressively smaller inhibition zones. *N. glabrata* H49 seemed to be the more resistant strain and *C. albicans* H37 the more sensitive strain to selected mouthwashes (Table 2).

#### 2.2.2. Growth Kinetics Assay

The results are available in the Appendix A. For all tested mouthwashes, a Minimum Fungicidal Concentration (MFC) was identified, as at least one of the three tested concentrations led to a 50% or greater reduction in optical density (OD) (Table 3). However, some unusual trends in the data may be explained by paradoxical growth or the release of excess biomass during co-incubation [20]. In some cases, lower concentrations of mouthwash resulted in higher OD values in growth kinetics, suggesting that fungal cells initially reacted differently depending on the concentration.

### 2.3. Biofilm Assays

#### 2.3.1. Minimal Biofilm Eradication Concentration

The CFU/mL inhibition levels showed highly satisfactory results for most of the tested mouthwashes, except for Arthrodont and Alodont, which were consistently less effective across all *Candida* species. *C. albicans* H37 exhibited the highest sensitivity, whereas *N. glabrata* ATCC2001 demonstrated the highest resistance profile to the tested mouthwashes (Figure 3).

#### 2.3.2. Biofilm Biomass Quantification

Crystal violet (CV) staining was used to evaluate the biomass production of *Candida* spp. biofilms after 24 h of contact with mouthwash (Table 4). Most mouthwashes effectively reduced biomass production, but Parodontax Extra and PerioPlus Curaprox showed the highest efficacy. This is likely due to their active ingredients—CHX 0.20% and CHX 0.20% + CPC, respectively. In contrast, Bexident Post 0.20%, despite also containing CHX 0.20%, had the lowest inhibitory effect on biomass production. This could be attributed to the presence of lactic acid, which creates an acidic environment that may indirectly support fungal growth. Notably, *C. albicans* MYK2760 exhibited the lowest biomass reduction, highlighting its higher resistance profile compared to other *Candida* species. Conversely, *N. glabrata* spp. showed the highest susceptibility, indicating greater sensitivity to these mouthwashes.

An increase in biomass—particularly noticeable in interactions between *C. albicans* or *N. glabrata* H49 and the mouthwash Bexident Post—has previously been documented in the context of antifungal treatments [21]. This phenomenon, often referred to as “paradoxical growth”, describes the unexpected ability of certain *Candida* species to resume growth at high concentrations of antifungal agents, rather than being inhibited. It is a well-recognized mechanism of resistance and is strongly associated with enhanced chitin biosynthesis in the fungal cell wall [22,23]. Similar responses have also been observed in reaction to certain mouthwashes, suggesting they may trigger comparable resistance mechanisms.

### 2.4. Morphological Alterations

The SEM images revealed that cell morphology is significantly affected by all the tested mouthwashes (Figure 4) in *C. albicans* SC5314. The images related to Parodontax suggest that some cells experience plasma membrane disruption, leading to intracellular material leakage and the formation of ring-like ghost structures (PX in SEM image). For Bexident Gums Daily use, Arthrodont, and Hexetidine, a notable alteration in membrane structure and a significant loss of cell density are observed compared to the control. Even though not all of them exhibit fungicidal effects, it can be inferred that Arthrodont, which showed the worst biofilm CFU results, likely has a fungistatic effect.

For the Vitis Gingival Mouthwash, zinc particles were detected and observed adhering to the cell surfaces (Figure 5A,C). This deposition could artificially increase the measured biomass, leading to a false positive in biomass quantification. Instead of actual fungal growth, the detected biomass may partially result from these zinc aggregates, potentially skewing the interpretation of the results.

## 3. Discussion

Mouthwashes contain various antimicrobial compounds that can influence the oral microbiota, including opportunistic fungi like *Candida* spp. Given the clinical relevance of oral candidiasis, evaluating the antifungal activity of these formulations is essential. Our findings highlight the antifungal efficacy of various tested mouthwashes, demonstrating that while most exhibit strong activity, their effects vary significantly between planktonic *Candida* cells and biofilms.

Among the tested mouthwashes, Listerine Cool Mint and Arthrodont were ineffective against planktonic *Candida* cells, while Arthrodont and Alodont failed to combat biofilms. Their formulations, primarily composed of natural compounds, may be less effective against *Candida* biofilms. Mouthwashes that inhibited *Candida* biofilms by more than two logs included those containing CHX alone, such as EluPerio and Parodontax Extra 0.20%; CHX combined with CPC, such as Gum Paroex 0.12%, Perio-Aid Intensive Care, and PerioPlus Curaprox; and CHX combined with a natural agent, such as Eludril Classic and Paroex 0.12%. Notably, the most unexpected yet intriguing result was observed with CPC combined with cinnamal in Oral-B Pro-Expert Professional Protection. CHX and CPC are well-known antimicrobial agents. Their mechanisms of action involve disruption of the fungal membrane, leading to the leakage of cytoplasmic components, interference with cellular metabolism, and inhibition of cell growth [24,25]. Similarly, cinnamaldehyde has been shown to have antifungal activity against *C. albicans* by inhibiting proteinase and phospholipase activities [26,27]. Additionally, the OD results for CV seem to suggest that dying fungal cells may release intracellular contents into the biofilm, which may potentially serve as nutrients for surviving *Candida* cells and contribute to persistent infections, increasing the overall OD, which has also been discussed [28,29].

When comparing our results with previous studies, it is important to note that the commercial name of a mouthwash does not always guarantee identical composition across products, which may explain certain discrepancies. Moreover, the majority of previous studies in the scientific literature overlook biofilm testing and focus on planktonic testing. Our findings align with those of Listerine’s antifungal activity against *C. albicans*, *N. glabrata*, and *C. parapsilosis* observed in the disc diffusion assay [30]. However, more recent studies have reported lower biofilm reduction rates for some mouthwashes compared to our results: for instance, Perio-Aid Intensive Care showed a 29.2% reduction in *C. albicans*, whereas our study demonstrated a significantly higher reduction of 99.99% in biofilms. Similarly, Gum Paroex exhibited a 29.2% reduction in *C. albicans* biofilms, while our findings showed a reduction of 99.9% in biofilms. Eludril Classic also demonstrated higher efficacy in our study, with a 99.9% reduction, compared to 41.7% of biofilms [31]. These may arise from the fact that there are differences in the assessment and the implemented methodologies to study the biofilms. A study reports a case of pseudomembranous oral candidiasis successfully treated with a mouthwash containing 0.05% CHX and 0.05% CPC [32]. This supports the potential of certain mouthwashes as therapeutic options for superficial oral fungal infections but highlights their limitation against deeper or systemic infections. These discrepancies highlight the importance of standardized methodologies and underscore the need for caution when interpreting results based solely on product labels or commercial claims.

Certain *N. glabrata* strains, such as ATCC 2001 and 15, exhibited heightened biofilm tolerance/resistance, consistent with the species’ well-documented resilience [33,34]. In planktonic assays, paradoxical growth was observed. This is a phenomenon observed in certain *Candida* species, where fungal growth is inhibited at lower antifungal concentrations but unexpectedly resumes at higher concentrations [20,35]. This effect is thought to result from adaptive stress responses or metabolic shifts that allow the cells to tolerate increased antimicrobial exposure [36]. In our study, this could explain why, in some cases, optical density measurements remained stable or even increased at higher mouthwash concentrations, rather than decreasing as typically expected. However, evidence of paradoxical growth in response to commonly used antiseptic agents such as CHX, CPC, benzoic acid, or fluoride compounds remains scarce. Studies consistently demonstrate that CHX effectively reduces both planktonic cells and also *C. albicans* biofilms without reported cases of paradoxical growth, suggesting that certain antiseptic formulations may bypass this adaptive resistance mechanism.

While mouthwashes have shown their in vitro effectiveness against *Candida* strains isolated from oral cavities, this clinical solution has certain limitations. One key constraint is that mouthwashes cannot be swallowed, restricting their action to the oral cavity and thereby limiting their efficacy in treating infections that extend to deeper mucosal tissues or systemic sites. Additionally, the formulation of some mouthwashes may influence their antifungal activity. For example, Parodontax Extra 0.20%, Eludril Classic, and Bexident Post 0.12% have a more lipophilic texture, which raises questions about how this property affects their ability to penetrate biofilms and exert antifungal effects. Another important consideration is the potential for drug combinations to improve antifungal outcomes. Combining azoles with antiseptics like CHX, CPC, or fluoride compounds may lead to synergistic or antagonistic interactions, potentially altering their effectiveness against *Candida* biofilms [37,38,39]. Additionally, since CHX, CPC, and fluoride compounds are already widely used in oral care, there is a concern that *Candida* species may have developed tolerance/resistance to these agents in real-life conditions, which could impact their clinical efficacy [40].

Our study encountered several limitations. While we aimed to analyse mouthwashes with diverse active ingredient compositions available in Europe, we faced significant constraints due to manufacturers’ lack of transparency regarding the concentrations of these ingredients. The absence of precise quantitative data introduces uncertainty into our analysis, limiting the formulation of statistically robust conclusions and reducing the precision of our efficacy assessments. Additionally, pharmaceutical excipients present in mouthwashes, such as emulsifiers and preservatives, can interact with active ingredients, potentially modifying their antifungal performance [24,41]. Lastly, comparing our results with those of other studies proved challenging due to variations in biofilm formation protocols. The lack of a standardized methodology for evaluating antifungal activity in *Candida* biofilms makes direct comparisons difficult and highlights the need for uniform testing procedures to ensure consistency across research studies [42].

## 4. Materials and Methods

### 4.1. Organisms and Growth Conditions

This study included reference and clinical isolates of *Candida* spp. for comparative analyses. Reference strains were tested, including *C. albicans* SC5314, as well as three from the American Type Culture Collection (ATCC, Manassas, VA, USA), *C. tropicalis* ATCC750, *N. glabrata* ATCC2001, and *C. parapsilosis* ATCC20019. Six clinical isolates were tested, including *C. albicans* MYK2760, *N. glabrata* H49, *N. glabrata* 15, *C. tropicalis* C7, *C. albicans* H37, and *C. albicans* H43 from our group’s mycotheque [14]. For each experiment, pre-inocula of *Candida* strains were cultured on Sabouraud dextrose agar (SDA) (Merck, Darmstadt, Germany) plates and incubated at 37 °C for 24 h.

### 4.2. Mouthwash Selection

Mouthwashes were selected according to inclusion and exclusion criteria. Mouthwashes that are commercially available in the European market, specifically target gum diseases, and have indications for adults were included. Products with other commercial presentations (gels or pastes) requiring a medical prescription were excluded. As a result of this selection, a total of 18 mouthwashes were evaluated in this study (Table 5), including Alodont (Recorcati, Warsaw, Poland), Arthrodont (Pierre Fabre, Castres, France), Bexident Gums Daily Use (Isdin, Barcelona, Spain), Bexident Intensive Gums 0.12% (Isdin, Barcelona, Spain), Bexident Post 0.20% (Isdin, Barcelona, Spain), Eludril Classic (Pierre Fabre, Castres, France), Eluperio (Pierre Fabre, Castres, France), Gum Gingidex 0.06% + CPC (Sunstar, Etoy, Switzerland), Gum Paroex 0.12% + CPC (Sunstar, Etoy, Switzerland), Hextril 0.10% (Johnson&Johnson, New Brunswick, NJ, USA), Listerine Cool Mint (Listerine, Morris Plains, NJ, USA), Oral-B Pro-Expert Professional Protection (Procter & Gamble, Cincinnati, OH, USA), Parodontax Extra 0.20% (Haleon, London, UK), Paroex 0.12% (Sunstar, Etoy, Switzerland), Perio·Aid Intensive Care (Dentaid, Barcelona, Spain), Perioplus Curaprox (Curaden, Kriens, Switzerland), Periogard (Colgate, New York, NY, USA), and Vitis Gingival Mouthwash (Dentaid, Barcelona, Spain). The mouthwashes were selected based on their active ingredients, including chlorhexidine digluconate (CHX), cetylpyridinium chloride (CPC), and essential oils. Each product was sourced directly from the manufacturers or authorized retailers.

### 4.3. Antifungal Susceptibility Testing (AFST)

#### 4.3.1. Disc Diffusion Assay

The antifungal susceptibility testing was conducted according to the EUCAST disc diffusion method [43]. A direct colony suspension method was used to prepare inocula in sterile saline, adjusted to the turbidity of a 0.5 McFarland standard. Three hundred microliters of the suspension were spread uniformly onto the surface of SDA plate using a sterile cotton swab. Inoculation was performed by swabbing the plates in three directions to ensure an even distribution of the inoculum, avoiding gaps between streaks. Paper discs of 6 mm diameter (PRAT DUMAS, Couze-St-Front, France) were placed onto the inoculated agar surface, and 10 µL of mouthwash (100% concentration) was applied to each disc. Positive controls were performed using chlorhexidine digluconate discs. Plates were incubated at 37 °C for 24 h on Sabouraud dextrose agar plates, and inhibition zones were measured to determine antifungal activity. This assay was performed in triplicate.

#### 4.3.2. Growth Kinetics in Planktonic *Candida* spp.

The determination of growth kinetics was obtained adapting the broth microdilution of *Candida* cells and mouthwashes, similar to the guidelines from [44]. Inocula were prepared by suspending five distinct colonies (≥1 mm in diameter) from 24 h cultures in at least 3 mL of sterile distilled water. The suspension was homogenized by vortexing for 15 s, and the cell density was adjusted to a 0.5 McFarland standard, with sterile distilled water added as necessary. A working suspension was then prepared by diluting the standardized suspension by 1:10 in sterile distilled water to achieve a final concentration of 1–5 × 10⁵ CFU/mL. The 96-well plate was inoculated as follows: for the negative control, 100 µL of the *Candida* strain suspension and 100 µL of RPMI-1640 medium (pH 7) supplemented with 2% glucose (Sigma-Aldrich, St. Louis, MO, USA); for the negative growth control, 200 µL of mouthwash. Test wells were prepared with 100 µL of mouthwash and 100 µL of *Candida* suspension (50% test), 50 µL of mouthwash, 50 µL of RPMI 1640, and 100 µL of *Candida* suspension (25% test), as well as 25 µL of mouthwash, 75 µL of RPMI 1640, and 100 µL of *Candida* suspension (12.5% test). Plates were incubated at 37 °C for 18–24 h. Results were visualized by measuring absorbance at 600 nm using a spectrophotometer. The assay was performed in triplicate.

### 4.4. Biofilm Assays

#### 4.4.1. Biofilm Reduction

Biofilm reduction was determined following an adaptation of the EUCAST method [14,44]. Around 2 to 4 colonies of *Candida* spp. were transferred to an Erlenmeyer flask containing 50 mL Sabouraud Dextrose Broth (SDB) (Merck, Darmstadt, Germany). The flask was incubated at 37 °C for 24 h at 120 rpm, with the lid loosened to allow gas dissipation. The optical density (OD) of the culture was adjusted to 0.1 at 530 nm using RPMI-1640 medium. Two hundred microliters of the adjusted cell suspension was inoculated into a 96-well plate and incubated at 37 °C for 24 h at 120 rpm. Following incubation, 100 µL of medium was removed from each well. Positive control wells received 100 µL of RPMI-1640, while test wells were treated with 100 µL of mouthwash (resulting in a 50% concentration of mouthwashes). The plate was incubated for an additional 6 h at 37 °C at 120 rpm. After co-incubation, the medium was removed, and each well was washed with 200 µL of phosphate-buffered saline (PBS; pH 7; 0.1 M) to eliminate planktonic cells. The PBS was then removed, and 200 µL of fresh PBS was added to each well. Biofilm cells were scraped from the well surfaces for suspension extraction. Twenty microliters of the biofilm suspension were transferred to a new 96-well plate containing 180 µL of PBS per well to prepare serial dilutions. Dilutions from 10^1^ to 10^7^ were performed by transferring 20 µL of suspension from one well to the next with thorough mixing (up-and-down pipetting) between each transfer. To enumerate colony-forming units per millilitre (CFU/mL), 10 µL of each dilution from 10^1^ to 10^7^ was spotted onto SDA plates. Drops were allowed to air dry before the plates were inverted and incubated at 37 °C for 24 h. After incubation, colonies were counted to determine CFU/mL for each dilution.

#### 4.4.2. Biofilm Biomass Quantification (Cristal Violet Assay)

*Candida* strains were fixed in the wells by removing all media and adding 200 µL of methanol for 15 min. The methanol was then removed, and the wells were allowed to dry completely. Biofilms were stained by adding 200 µL of 10% crystal violet (CV) solution for 5 min. Excess CV was removed, and the wells were washed twice with deionized water. After removing the deionized water, the wells were dried thoroughly. To solubilize the dye bound to the biofilm matrix, 200 µL of 33% acetic acid was added to each well and mixed thoroughly. The absorbance of the resulting solution was measured at 570 nm using a spectrophotometer [45]. Each mouthwash was used as a blank to account for any potential interference from residual components in the OD measurements. The assay was performed in triplicate.

### 4.5. Morphological Alterations

In order to examine the potential morphological alterations of the biofilms of *C. albicans* SC5314, scanning electron microscopy (SEM) was performed at the Scanning Electron Microscopy Laboratory of CEMUP (Materials Centre of the University of Porto, Portugal). Samples were sequentially treated with 70% ethanol for 10 min, 95% ethanol for 10 min, and 100% ethanol for 20 min. Following dehydration, samples were placed in a desiccator to dry completely. The SEM examination was conducted using a high-resolution (Schottky) Environmental Scanning Electron Microscope (FEI Quanta 400 FEG ESEM) equipped with X-ray microanalysis and Backscattered Electron (BSE) diffraction analysis (EDAX Genesis X4M). Prior to imaging, the samples were coated with a thin Au/Pd film via sputtering using an SPI Module Sputter Coater [45]. Observations were carried out using Secondary Electrons (SEs) at an accelerating voltage of 10 keV with magnifications of 500×, 2500×, and 10,000×. Backscattered Electron (BSE) diffraction imaging was performed at 10 keV and 15 keV with magnifications of 10,000× and 50,000×.

## 5. Statistical Analysis

All measurements and analyses were carried out, at least two times, in triplicate. The experimental data were evaluated by GraphPad Prism v.9.1.1 software (San Diego, CA, USA). In all cases, statistical significance was set as *p* < 0.05. Data are presented as the mean ± standard deviation (SD).

## 6. Conclusions

Several of the tested mouthwashes demonstrated potent antifungal activity against *Candida* spp. Notably, mouthwashes containing CHX, either alone or in combination with CPC and other active compounds, emerged as a common factor among the most efficacious formulations. Additionally, cinnamaldehyde combined with CPC exhibited promising activity against *Candida* biofilms. The occurrence of paradoxical effects may provide new insights into the antifungal mechanisms of these compounds and warrant further investigation into their potential therapeutic applications for oral candidiasis. To conclude, rather than replacing conventional antifungal therapies, these solutions can serve as complementary strategies in the current effort to combat antifungal resistance, which is why more studies with standardized methodologies are suggested.

## Figures and Tables

**Figure 1 antibiotics-14-00483-f001:**
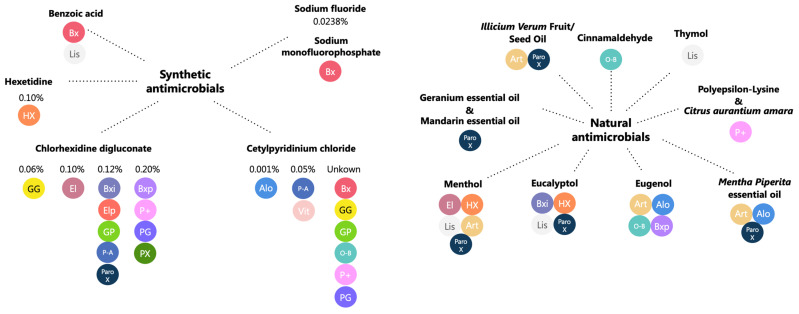
Antimicrobial substances contained in mouthwashes formulations with their respective concentrations. Abbreviations—Alo: Alodont; Art: Arthrodont; Bx: Bexident Gums Daily use; Bxi: Bexident intensive gums 0.12%; Bxp: Bexident Post 0.20%; El: Eludril Classic; Elp: EluPerio; GG: Gum Gingidex 0.06% + CPC; GP: Gum Paroex 0.12% + CPC; Hx: Hextril 0.10%; Lis: Listerine Cool Mint; O-B: Oral-B Pro-expert Professional Protection; PX: Parodontax Extra 0.20%; ParoX: Paroex 0.12%; P-A: Perio-Aid Intensive Care; P+: PerioPlus Curaprox; PG: Periogard; Vit: Vitis Gingival Mouthwash.

**Figure 2 antibiotics-14-00483-f002:**
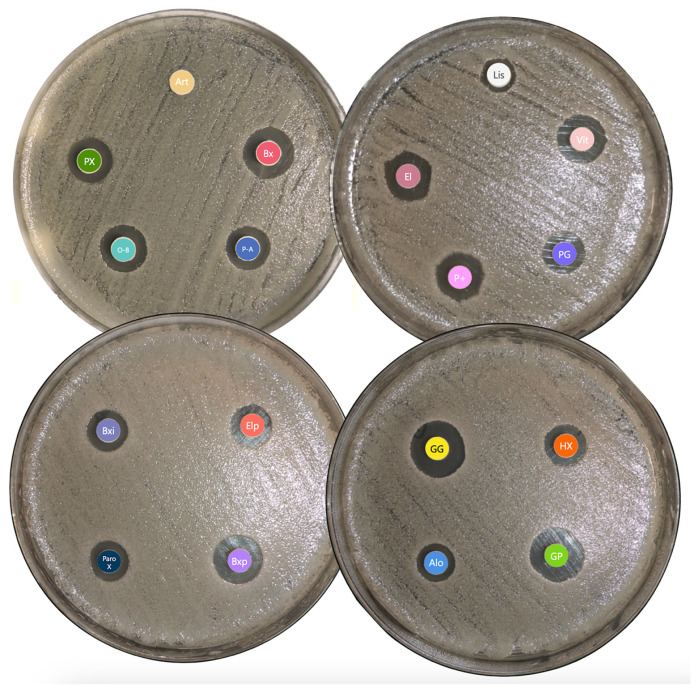
Disc Diffusion assay results for *C. albicans* SC5314. Abbreviations—Alo: Alodont; Art: Arthrodont; Bx: Bexident Gums Daily use; Bxi: Bexident intensive gums 0.12%; Bxp: Bexident Post 0.20%; El: Eludril Classic; Elp: EluPerio; GG: Gum Gingidex 0.06% + CPC; GP: Gum Paroex 0.12% + CPC; Hx: Hextril 0.10%; Lis: Listerine Cool Mint; O-B: Oral-B Pro-expert Professional Protection; PX: Parodontax Extra 0.20%; ParoX: Paroex 0.12%; P-A: Perio-Aid Intensive Care; P+: PerioPlus Curaprox; PG: Periogard; Vit: Vitis Gingival Mouthwash.

**Figure 3 antibiotics-14-00483-f003:**
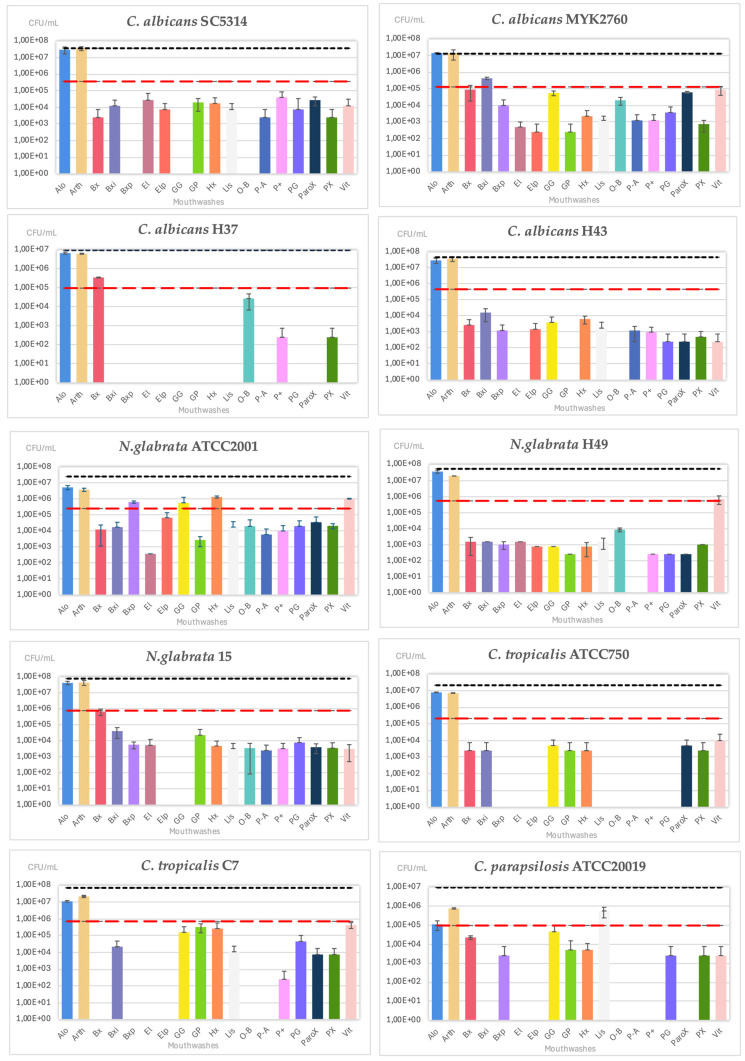
Biofilm quantification in CFU/mL of *Candida* strains exposed to selected mouthwashes. The black line is the positive control, and the red line stands for a reduction of 99% (MBEC).

**Figure 4 antibiotics-14-00483-f004:**
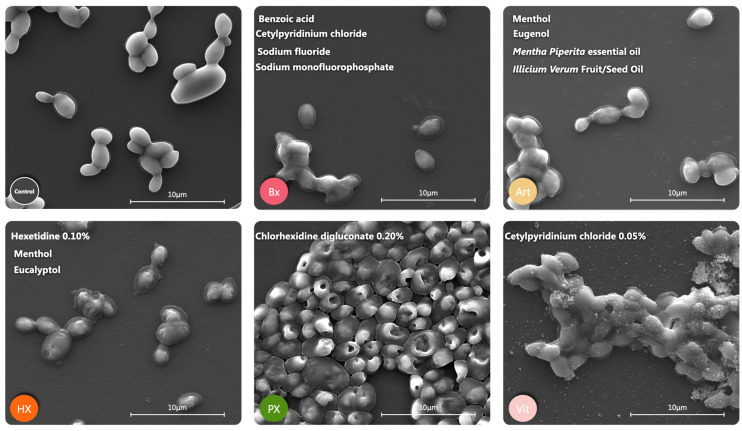
SEM image of morphological alteration of on *C. albicans* SC 5314 by Secondary Electrons (SEs) at 10,000× magnification and 10 kV.

**Figure 5 antibiotics-14-00483-f005:**
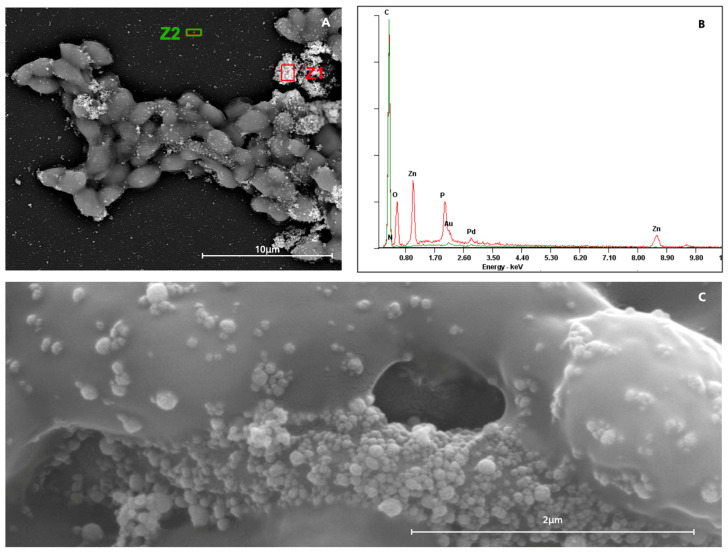
SEM analysis of Vitis Gingival Mouthwash in *C. albicans* SC 5314. (**A**) Composition alteration by Backscattered Electrons (BSEs) at 15 keV. (**B**) Composition comparison between Z1 (red line) and Z2 (green line) zones at 15 keV. (**C**) Morphological alteration by Secondary Electrons (SEs) at 50,000× magnification and 10 kV.

**Table 1 antibiotics-14-00483-t001:** Studied mouthwashes composition (numbers in %).

	MW	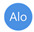	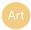	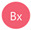	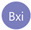	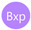	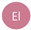	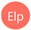	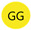	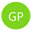	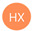	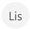	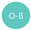	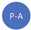	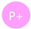	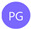	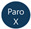	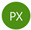	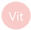	TOTAL
Component	
Alcohol	+	+				+					+								22%
Allantoin			+	+	+													+	22%
Anethol																+			6%
Aroma		+	+	+	+	+	+	+	+		+	+	+	+	+	+	+	+	89%
Benzoic acid			+								+								11%
Benzyl alcohol							+												6%
Calcium acetate sodium										+									6%
Chitosan					+														6%
Chlorobutanol	0.01					0.5													6%
CHX				0.12	0.2	0.1	0.12	0.06	0.12				0.12	0.2	0.2	0.12	0.2		61%
CI 14720 (Azorubine)									+	+						+			17%
CI 16035																		+	6%
CI 16255						+	+												11%
CI 42051	+	+													+				17%
CI 42053											+								6%
CI 42090												+	+						11%
CI 47005											+								6%
Cinnamal												+							6%
Citric acid	+		+	+				+	+	+				+	+				44%
Citrus aurantium amara														+					6%
Cocamidopropyl Betaine				+															6%
CPC	0.001		+					+	+			+	0.05	+	+			0.05	39%
D-Panthenol			+	+	+													+	22%
Diethylhexyl sodium sulfosuccinate						+													6%
Dipotassium Glycyrrhizate		+																	6%
Ethanol										96						+			11%
Eucalyptol				+						+	+					+			22%
Eugenol	0.001	+			+							+							17%
Gellan Gum																	+		6%
*Geranium* essential oil																+			6%
Glycerin/Glycerol		+				+	+					+	+	+	+	+		+	50%
Hexetidine										0.1									6%
Hydroxyethyl cellulose					+														6%
*Illicium verum* Fruit/Seed Oil		+														+			11%
Lactic acid		+			+													+	17%
Laureth-9		+																	6%
Limonene		+	+	+	+	+	+												33%
Maltol																+			6%
Mandarin essential oil																+			6%
*Mentha piperita* oil	+	+														+			17%
Menthol		+				+				+	+					+			28%
Menthone																+			6%
Menthyl acetate																+			6%
Methylparaben												+						+	11%
Metyl salicylate											+								6%
Neohesperin Dihydrchalone													+						6%
o-cymen-5-ol			+																6%
PEG-25	+																		6%
PEG-40			+	+	+		+	+	+				+		+	+			50%
PEG-60 hydrogenated castor oil		+															+		11%
Phenoxyethanol														+					6%
Poloxamer 188				+															6%
Poloxamer 407											+	+						+	17%
Polyepsilon-Lysine														+					6%
Polysorbate 20														+					6%
Polysorbate 80										+									6%
Potassium acesulfame							+						+						11%
Potassium nitrate																			0%
Potassium sorbate			+																6%
Propolys extracts																			0%
Propylene Glycol			+				+	+	+				+		+	+		+	44%
Propylparaben												+							6%
Sodium benzoate		+		+							+							+	22%
Sodium chloride														+					6%
Sodium citrate			+	+				+	+										22%
Sodium fluoride			0.024								0.03								11%
Sodium gluconate																		+	6%
Sodium hydroxyide	+									+				+					17%
Sodium lactate																		+	6%
Sodium monofluorophosphate			+																6%
Sodium saccharin	+		+	+	+					+	+	+	+					+	50%
Sorbitol			+	+	+						+				+		+		33%
Sucralose								+	+					+		+			22%
Thymol											+								6%
Triacetin																+			6%
Vaniline																+			6%
VP/VA Copolymer														+					6%
Water	+	+	+	+	+	+	+	+	+	+	+	+	+	+	+	+	+	+	100%
Xylitol													+	+				+	17%
Zinc Chloride											+								6%
Zinc Lactate																		+	6%

The sign “+” stands for “present in the composition”. Abbreviations—Alo: Alodont; Art: Arthrodont; Bx: Bexident Gums Daily use; Bxi: Bexident Intensive gums 0.12%; Bxp: Bexident Post 0.20%; El: Eludril Classic; Elp: EluPerio; GG: Gum Gingidex 0.06% + CPC; GP: Gum Paroex 0.12% + CPC; Hx: Hextril 0.10%; Lis: Listerine Cool Mint; MW: Mouthwash; O-B: Oral-B Pro-expert Professional Protection; PX: Parodontax Extra 0.20%; ParoX: Paroex 0.12%; P-A: Perio-Aid Intensive Care; P+: PerioPlus Curaprox; PG: Periogard; Vit: Vitis Gingival Mouthwash.

**Table 2 antibiotics-14-00483-t002:** Inhibition zone (in mm) obtained in disc diffusion assay.

	Strains	*C. albicans* SC 5314	*C. albicans* MYK2760	*C. albicans* H37	*C. albicans* H43	*N. glabrata* ATCC2001	*N. glabrata* H49	*N. glabrata* 15	*C. tropicalis* ATCC750	*C. tropicalis* C7	*C. parapsilosis* ATCC 20019
MW	
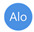	2.25 ± 0.35	0 ± 0.0	0 ± 0.0	3.75 ± 0.35	3.75 ± 0.35	0 ± 0.0	3.5 ± 0.71	1.5 ± 0.71	0 ± 0.0	0.25 ± 0.35
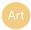	0 ± 0.0	0 ± 0.0	0 ± 0.0	0 ± 0.0	0 ± 0.0	0 ± 0.0	0 ± 0.0	0 ± 0.0	0.5 ± 0.71	0 ± 0.0
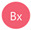	6 ± 1.41	4 ± 0.0	4.25 ± 0.35	6.75 ± 0.35	6.75 ± 1.77	0 ± 0.0	6.75 ± 0.35	4.5 ± 0.71	4 ± 0.0	4.25 ± 0.35
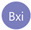	3.25 ± 0.35	6 ± 0.0	3 ± 0.71	3.25 ± 0.35	4 ± 0.71	8.25 ± 1.77	4 ± 0.0	7.75 ± 1.06	7 ± 0	5.5 ± 0.0
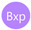	6 ± 0.0	6.5 ± 2.12	5.75 ± 1.06	5.75 ± 1.77	5 ± 1.41	9.75 ± 1.77	5.25 ± 0.35	7.75 ± 1.06	9.5 ± 2.12	6 ± 0.71
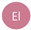	3.75 ± 2.47	5 ± 1.41	5.75 ± 1.77	7 ± 5.66	4 ± 1.41	10.75 ± 3.89	4 ± 1.41	10 ± 7.07	11 ± 2.83	6 ± 1.41
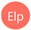	4 ± 0.0	5.75 ± 0.35	4 ± 0.0	3.25 ± 0.35	2.75 ± 0.35	9.75 ± 0.35	3.5 ± 0.71	8.5 ± 0.71	7.5 ± 0.71	4 ± 0.0
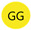	7.75 ± 0.35	3.75 ± 0.35	4 ± 0.71	7.5 ± 1.41	7 ± 0.71	7.75 ± 0.35	6.75 ± 0.35	7.25 ± 0.35	5 ± 1.41	5.75 ± 0.35
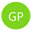	4.5 ± 3.54	4.5 ± 2.12	3.5 ± 0.71	7 ± 0	8.5 ± 0.71	10.5 ± 0.71	8.5 ± 0.71	8.75 ± 0.35	7.75 ± 1.06	7.75 ± 0.35
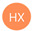	1.75 ± 2.47	4 ± 1.41	3.5 ± 0.71	0.25 ± 0.35	2 ± 1.41	5.5 ± 2.83	4.75 ± 1.06	1 ± 0.0	3 ± 1.41	1.75 ± 1.77
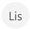	1 ± 0.0	0.5 ± 0.71	1 ± 0.0	2 ± 0.71	0.75 ± 0.35	0.75 ± 0.35	1 ± 0.0	0 ± 0.0	0.5 ± 0.71	0 ± 0.0
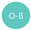	5.5 ± 1.41	3 ± 0.0	3.5 ± 0.71	6 ± 1.41	5.5 ± 0.71	1.5 ± 0.71	6 ± 1.41	5 ± 1.41	3.75 ± 0.35	6 ± 0.0
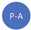	5.5 ± 2.12	2.75 ± 0.35	2.25 ± 1.06	4.25 ± 0.35	3.75 ± 0.35	7 ± 1.41	6 ± 0.0	6 ± 1.41	5.75 ± 0.35	5.75 ± 1.06
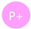	6 ± 0.0	5.25 ± 1.77	4.5 ± 2.12	6.25 ± 1.06	5.75 ± 1.77	10 ± 2.83	6 ± 1.41	8.5 ± 0.71	7.25 ± 1.06	6.75 ± 1.06
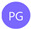	4.75 ± 0.35	7.25 ± 1.06	6.25 ± 1.06	4.25 ± 0.35	5 ± 1.41	10.5 ± 2.12	5.75 ± 1.06	9 ± 1.41	8.5 ± 3.55	5.75 ± 0.35
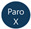	3.75 ± 0.35	5.75 ± 0.35	4.75 ± 0.35	3.75 ± 0.35	4 ± 1.41	9.25 ± 1.06	3.75 ± 0.35	9.25 ± 0.35	7.5 ± 0.71	5.25 ± 1.06
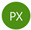	4.25 ± 0.35	6.75 ± 1.06	6.5 ± 0.71	4.5 ± 0.71	4.5 ± 0.71	9.5 ± 1.41	4 ± 2.83	8.5 ± 0.71	7 ± 0	6.5 ± 0.71
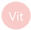	6 ± 0.71	4.25 ± 1.06	4.5 ± 0.71	6 ± 1.41	5.75 ± 1.06	1.5 ± 0.71	5.75 ± 1.77	6 ± 1.41	3.75 ± 0.35	5 ± 1.41

Note: The strongest inhibitions are marked in green, while the weakest are in red. The white and intermediates tonalities stand for intercalary values. Abbreviations—Alo: Alodont; Art: Arthrodont; Bx: Bexident Gums Daily use; Bxi: Bexident Intensive gums 0.12%; Bxp: Bexident Post 0.20%; El: Eludril Classic; Elp: EluPerio; GG: Gum Gingidex 0.06% + CPC; GP: Gum Paroex 0.12% + CPC; Hx: Hextril 0.10%; Lis: Listerine Cool Mint; MW: Mouthwash; O-B: Oral-B Pro-expert Professional Protection; PX: Parodontax Extra 0.20%; ParoX: Paroex 0.12%; P-A: Perio-Aid Intensive Care; P+: PerioPlus Curaprox; PG: Periogard; Vit: Vitis Gingival Mouthwash.

**Table 3 antibiotics-14-00483-t003:** Growth kinetics of *C. albicans* SC5314.

	MouthwashConcentration	MFC	IncreasedGrowth
	12.5%	25%	50%
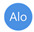	9.6%	4.4%	**−0.9%**	50%	No
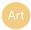	119.0%	91.7%	**3.9%**	50%	No
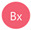	**5.1%**	6.0%	14.3%	12.5%	**Yes**
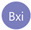	**10.4%**	10.6%	14.2%	12.5%	**Yes**
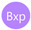	**6.6%**	8.4%	6.7%	12.5%	**Yes**
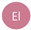	**−5.2%**	11.6%	124.9%	12.5%	**Yes**
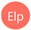	**−11.9%**	−4.2%	1.4%	12.5%	**Yes**
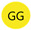	6.1%	4.5%	**1.7%**	50%	No
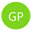	**−2.6%**	0.4%	13.7%	12.5%	**Yes**
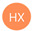	**−15.8%**	−8.6%	6.8%	12.5%	**Yes**
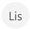	55.3%	9.9%	**1.6%**	50%	No
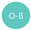	4.9%	5.4%	**2.3%**	50%	No
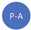	3.2%	2.8%	**−1.7%**	50%	No
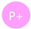	5.3%	4.2%	**−0.3%**	50%	No
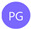	4.2%	**2.9%**	4.9%	25%	**Yes**
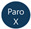	**−8.2%**	−4.9%	−1.1%	12.5%	**Yes**
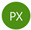	8.3%	6.1%	**5.5%**	50%	No
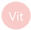	5.6%	7.1%	**0.9%**	50%	No

**Table 4 antibiotics-14-00483-t004:** Biomass reduction by CV measurement (in %).

	Strains	*C. albicans* SC 5314	*C. albicans* MYK2760	*C. albicans* H37	*C. albicans* H43	*N. glabrata* ATCC2001	*N. glabrata* H49	*N. glabrata* 15	*C. tropicalis* ATCC750	*C. tropicalis* C7	*C. parapsilosis* ATCC 20019
MW	
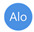	9%	−152%	−147%	3%	52%	−126%	24%	−14%	15%	−39%
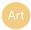	−3%	−82%	−54%	23%	36%	−158%	71%	17%	24%	−48%
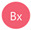	31%	−13%	33%	65%	64%	15%	81%	25%	24%	36%
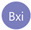	24%	−32%	50%	52%	58%	25%	62%	5%	32%	47%
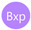	−78%	−263%	−112%	−17%	28%	−200%	43%	−6%	−115%	−35%
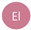	90%	−16%	20%	59%	11%	30%	89%	42%	−26%	49%
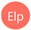	32%	−25%	−4%	54%	63%	37%	35%	37%	34%	39%
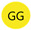	51%	−29%	−17%	76%	80%	16%	46%	35%	31%	45%
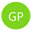	50%	−72%	−11%	67%	78%	37%	59%	29%	18%	28%
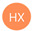	62%	−2%	−40%	50%	80%	48%	76%	−42%	6%	−49%
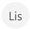	11%	26%	74%	30%	58%	57%	67%	−29%	64%	18%
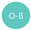	31%	3%	32%	62%	76%	−31%	46%	−31%	−61%	−41%
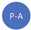	49%	39%	45%	50%	64%	38%	49%	1%	−3%	−63%
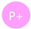	58%	44%	54%	59%	55%	44%	73%	1%	−3%	6%
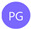	52%	39%	5%	64%	32%	53%	68%	9%	25%	−47%
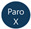	50%	28%	4%	62%	66%	63%	55%	2%	18%	2%
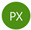	61%	16%	52%	95%	74%	64%	77%	20%	14%	−1%
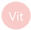	18%	2%	2%	−24%	44%	29%	63%	−16%	10%	−23%

Note: The strongest inhibitions are marked in green, while the weakest are in red. The white and intermediates tonalities stand for intercalary values. Abbreviations—Alo: Alodont; Art: Arthrodont; Bx: Bexident Gums Daily use; Bxi: Bexident Intensive gums 0.12%; Bxp: Bexident Post 0.20%; El: Eludril Classic; Elp: EluPerio; GG: Gum Gingidex 0.06% + CPC; GP: Gum Paroex 0.12% + CPC; Hx: Hextril 0.10%; Lis: Listerine Cool Mint; MW: Mouthwash; O-B: Oral-B Pro-expert Professional Protection; PX: Parodontax Extra 0.20%; ParoX: Paroex 0.12%; P-A: Perio-Aid Intensive Care; P+: PerioPlus Curaprox; PG: Periogard; Vit: Vitis Gingival Mouthwash.

**Table 5 antibiotics-14-00483-t005:** Studied mouthwashes and their respective symbol.

Symbol	Mouthwash
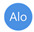	Alodont
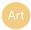	Arthrodont
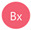	Bexident Gums Daily use
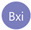	Bexident Intensive gums 0.12%
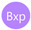	Bexident Post 0.20%
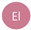	Eludril Classic
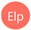	Eluperio
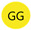	Gum Gingidex 0.06% + CPC
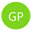	Gum Paroex 0.12% + CPC
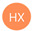	Hextril 0.10%
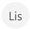	Listerine Cool Mint
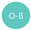	Oral-B Pro-expert Professional Proctection
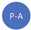	Parodontax Extra 0.20%
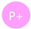	Paroex 0.12%
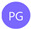	Perioplus Curaprox
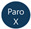	Perio·Aid Intensive Care
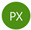	Periogard
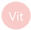	Vitis Gingival Mouthwash

## Data Availability

The original contributions presented in this study are included in the article. Further inquiries can be directed to the corresponding author.

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
