# Peer review of "Repurposing Mouthwashes: Antifungal and Antibiofilm Abilities of Commercially Available Mouthwashes Against Candida spp."

_antibiotics, 2025, doi:10.3390/antibiotics14050483_

Round 1
Reviewer 1 Report
Comments and Suggestions for Authors
The article offers a thorough and insightful exploration of how mouthwashes affect Candida, particularly in terms of their antifungal activity at different concentrations and interactions with biofilms. The manuscript is well-organized, and the methods used are solid, based on reliable protocols like the EUCAST disc diffusion method. The research provides meaningful contributions to the field, and the approach of studying mouthwash formulations against Candida biofilms is especially innovative.
The introduction effectively introduces the research question, but one suggestion is to clarify the part about optical density measurements. The sentence describing the optical density measurement could be clearer by replacing "did not decrease as expected" with a more precise explanation of what was observed experimentally. Providing a more detailed explanation would make the text easier to follow for a broader audience. Also, briefly explaining "paradoxical growth" would help readers who are not familiar with this concept, though it's not a major issue for those already acquainted with it.
The section on CHX efficacy is well-written, with clear details about its benefits. A small tweak to the sentence about its "efficacy against biofilms and planktonic forms of C. albicans" could improve the flow of the paragraph, but this is a minor adjustment in an otherwise solid section.
The discussion on the limitations of mouthwashes is good, though it could be made even stronger by more explicitly mentioning the challenges of treating deeper infections. This would give a little more depth to the analysis, but the current explanation is still quite effective.
The methodology is well-detailed and clear, with biofilm assays presented in a straightforward way. A bit more detail on variables like mouthwash concentration and incubation time could make the experiments even more reproducible, but overall, the approach is solid and reliable.
In conclusion, the article is well-structured and makes a valuable contribution to the field. The suggested changes are minor, but if addressed, they would further improve the clarity and scientific rigor of the manuscript. The study of mouthwash formulations against Candida biofilms is a particularly innovative aspect, and the article lays a strong foundation for future research in this area.

Author Response
Dear Reviewer,
Thank you very much for taking the time to review our manuscript. Your feedback and suggestions are greatly appreciated. We have carefully considered your comments and have made the necessary revisions accordingly. We are grateful for the opportunity to address your concerns and hope that the revised version meets your expectations. Please let us know if there are any further adjustments or clarifications needed. Once again, thank you for your valuable input and for helping us improve our work.
Comment 1: The sentence describing the optical density measurement could be clearer by replacing "did not decrease as expected" with a more precise explanation of what was observed experimentally.
Response 1 : We rephrased “The optical density measurements remained stable or even increased at higher mouthwash concentrations, rather than decreasing as typically expected”.
Comment 2: Providing a more detailed explanation would make the text easier to follow for a broader audience. Also, briefly explaining "paradoxical growth" would help readers who are not familiar with this concept, though it's not a major issue for those already acquainted with it.
Response 2 : It was explained line 235-239 “This is a phenomenon observed in certain Candida species, where fungal growth is inhibited at lower antifungal concentrations but unexpectedly resumes at higher concentrations. This effect is thought to result from adaptive stress responses or metabolic shifts that allow the cells to tolerate increased antimicrobial exposure’.
Comment 3: A small tweak to the sentence about its "efficacy against biofilms and planktonic forms of C. albicans" could improve the flow of the paragraph, but this is a minor adjustment in an otherwise solid section.
Response 3 : Thank you. We changed it to: "efficacy against planktonic cells and also in C. albicans biofilms". We hope that this is clearer now.
Comment 4: The discussion on the limitations of mouthwashes is good, though it could be made even stronger by more explicitly mentioning the challenges of treating deeper infections. This would give a little more depth to the analysis, but the current explanation is still quite effective.
Response 4 : We appreciate the reviewer’s feedback and have clarified the limitation more explicitly in line 248. The sentence now reads: “One key constraint is that mouthwashes cannot be swallowed, restricting their action to the oral cavity and thereby limiting their efficacy in treating infections that extend to deeper mucosal tissues or systemic sites.”
Comment 5: The methodology is well-detailed and clear, with biofilm assays presented in a straightforward way. A bit more detail on variables like mouthwash concentration and incubation time could make the experiments even more reproducible, but overall, the approach is solid and reliable.
Response 5 : In response to the suggestion for greater detail, we have clarified the mouthwash concentrations by adding “100% concentration” in line 315 and specifying “resulting in a 50% concentration of mouthwashes” in line 344. Incubation times are consistently reported throughout the methods section to ensure reproducibility.
Reviewer 2 Report
Comments and Suggestions for Authors
The study evaluated in vitro antifungal efficacy of 18 mouthwashes against Candida spp., identifying Chlorhexidine-based formulations as the most effective, with potential as adjuncts in the treatment of oral candidiasis. However, the study was limited by the lack of precise data on active ingredient concentrations and the potential interactions between excipients and active compounds in the mouthwashes. Besides these issues, there are several areas where the manuscript could be improved. Here are some comments for the authors to consider:
Abstract
- Please revise the opening sentence to eliminate colloquial expressions such as "what if the solution to...". The abstract should maintain a formal and scientific tone, avoiding rhetorical questions.
Introduction
- The examples of repurposed drugs for Parkinson’s and Alzheimer’s are interesting but not clearly tied to antifungal or antimicrobial contexts. Consider referencing antifungal-specific repurposing examples (e.g., NSAIDs, or anticancer drugs with known antifungal activity) if available, to maintain thematic focus.
Material and method
- No mention of ethics approval or patient consent if clinical isolates were derived from identifiable sources.
- In Antifungal Susceptibility Testing (AFST), there is no mention of negative controls (e.g., saline or blank disc) or positive antifungal controls (e.g., fluconazole disc). Were assays performed in triplicate? Include details on how variability across replicates was handled statistically.
- In the Growth Kinetics section, control definitions need refinement. The use of “positive control” to describe “mouthwash only” is misleading, as no Candidais present in that condition. Please revise to “negative growth control.”
- For the Biofilm Biomass (Crystal Violet) Assay, state whether mouthwash residues interfered with OD readings, and if so, how this was corrected or accounted for.
- In the SEM Imaging section, essential imaging parameters are missing. Please specify the magnification, resolution, voltage, and any methods used for image analysis or quantification.
Results
- There is no information on the number of biological repeats or technical replicates performed for any assay in manuscript.
- The absence of statistical analysis (e.g., p-values, error bars, standard deviation) throughout the manuscript significantly undermines the credibility of the data. Please include appropriate statistical treatment throughout the manuscript.
- In section 2.2.2, it would be nice to include growth kinetics of SC5314 in the main manuscript and other strains growth kinetics could be include in Supplementary table.
- In Figure 3, please clarify which threshold denotes MBEC (e.g., >2 log10 reduction = MBEC?). The figure legend should specify this.
- The X- and Y-axis labels are missing in Figure 3.
- In table 3,several results show negative biomass reductions (e.g., -263%) which implies increased biofilm mass. This is not discussed at all.
- When referring to "ring-like ghost structures" or "zinc particles," the corresponding figure panels (a, b, c) should be explicitly referenced for clarity.
- While the qualitative SEM analysis is well discussed, it would strengthen the conclusions if quantitative metrics—such as cell wall thickness, pore size, or percentage of disrupted cells—were included, if available.
- The text inconsistently refers to “Hexitidine”, please change in “Hexetidine” throughout the manuscript.
Author Response
Dear Reviewer,
Thank you very much for taking the time to review our manuscript. Your feedback and suggestions are greatly appreciated. We have carefully considered your comments and have made the necessary revisions accordingly. We are grateful for the opportunity to address your concerns and hope that the revised version meets your expectations. Please let us know if there are any further adjustments or clarifications needed. Once again, thank you for your valuable input and for helping us improve our work.
Comment 1: Abstract : Please revise the opening sentence to eliminate colloquial expressions such as "what if the solution to...". The abstract should maintain a formal and scientific tone, avoiding rhetorical questions.
Response 1 : We have refined the sentence in order to maintain the scientific tone : “Indeed, the solution to emerging infectious diseases may no longer lie in costly new drug development, but rather in unlocking the untapped potential of existing substances”
Comment 2: Introduction: The examples of repurposed drugs for Parkinson’s and Alzheimer’s are interesting but not clearly tied to antifungal or antimicrobial contexts. Consider referencing antifungal-specific repurposing examples (e.g., NSAIDs, or anticancer drugs with known antifungal activity) if available, to maintain thematic focus.
Response 2 : We appreciate the reviewer’s suggestion and agree that providing antifungal-specific repurposing examples would enhance the thematic consistency: “Recent studies have explored the antifungal properties of non-steroidal anti-inflammatory drugs (NSAIDs) like ibuprofen and diclofenac, which have shown inhibitory effects against Candida species and other fungal pathogens [3,4]. Additionally, anticancer drugs, such as tamoxifen and bleomycin, have demonstrated antifungal activity, suggesting that their repurposing could provide alternative treatment options for fungal infections [5,6].”
Comment 3: Material and method : No mention of ethics approval or patient consent if clinical isolates were derived from identifiable sources.
Response 3 : The collection of clinical isolates was performed in another paper : Alves, A.M.C.V.; Lopes, B.O.; Leite, A.C.R. de M.; Cruz, G.S.; Brito, É.H.S. de; Lima, L.F. de; ÄŒernáková, L.; Azevedo, N.F.; Rodrigues, C.F. Characterization of Oral Candida Spp. Biofilms in Children and Adults Carriers from Eastern Europe and South America. Antibiotics 2023, 12, 797, doi:10.3390/antibiotics12050797. : “The study was preceded by the approval of the committee of ethics in research (CEP) under the number 4.432.501, following the ethical aspects of the resolution 466/12 and 510/16 of the Conselho Nacional de Saúde”
Comment 4: Material and method : In Antifungal Susceptibility Testing (AFST), there is no mention of negative controls (e.g., saline or blank disc) or positive antifungal controls (e.g., fluconazole disc). Were assays performed in triplicate? Include details on how variability across replicates was handled statistically.
Response 4 : Thank you for your comment. We added the missing information: "positive controls were performed using chlorhexidine discs". We also added “The assay was performed in triplicate”
Comment 5: Material and method : In the Growth Kinetics section, control definitions need refinement. The use of “positive control” to describe “mouthwash only” is misleading, as no Candida is present in that condition. Please revise to “negative growth control.”
Response 5 : Confirmed — the modification is completed.
Comment 6: Material and method : For the Biofilm Biomass (Crystal Violet) Assay, state whether mouthwash residues interfered with OD readings, and if so, how this was corrected or accounted for.
Response 6 : We used each mouthwash as a blank control to ensure that residues did not interfere with the OD readings. To clarify we add : “Each mouthwash was used as a blank to account for any potential interference from residual components in the OD measurements”
Comment 7: Material and method : In the SEM Imaging section, essential imaging parameters are missing. Please specify the magnification, resolution, voltage, and any methods used for image analysis or quantification.
Response 7 : We added missing parameters : “The SEM examination was conducted using a high-resolution (Schottky) Environmental Scanning Electron Microscope (FEI Quanta 400 FEG ESEM) equipped with X-ray mi-croanalysis and Electron Backscattered Diffraction (BSE) analysis (EDAX Genesis X4M). Prior to imaging, the samples were coated with a thin Au/Pd film via sputtering using an SPI Module Sputter Coater [42]. Observations were carried out using Secondary Electrons (SE) at an accelerating voltage of 10 keV with magnifications of 500×, 2,500×, and 10,000×. Electron Backscattered Diffraction (BSE) imaging was performed at 10 keV and 15 keV with magnifications of 10,000× and 50,000×.”
Comment 8: Results: There is no information on the number of biological repeats or technical replicates performed for any assay in manuscript.
Response 8: thank you for your important comment. Indeed, this information was not included in the original manuscript. Therefore, we have added the following sentence: 'The assay was performed in triplicate.'
Comment 9: Results: The absence of statistical analysis (e.g., p-values, error bars, standard deviation) throughout the manuscript significantly undermines the credibility of the data. Please include appropriate statistical treatment throughout the manuscript.
Response 9: Thank you for your comment. We apologize for this fault. We added SDs in Disc diffusion assay and error bars in Figure 3. Biofilm quantification in CFU/mL
Comment 10: Results: In section 2.2.2, it would be nice to include growth kinetics of SC5314 in the main manuscript and other strains growth kinetics could be include in Supplementary table.
Response 10: Done.
Comment 11: Results: In Figure 3, please clarify which threshold denotes MBEC (e.g., >2 log10 reduction = MBEC?). The figure legend should specify this.
Response 11: Thank you for pointing this out in figure 3. Now the legend reads : “The black line is the positive control, and red line stands for a reduction of 99% (MBEC).”
Comment 12: Results: The X- and Y-axis labels are missing in Figure 3.
Response 12 : Done.
Comment 13: Results: In table 3,several results show negative biomass reductions (e.g., -263%) which implies increased biofilm mass. This is not discussed at all.
Response 13 : Thank you for your comment. We added this paragraph : “An increase in biomass—particularly noticeable in interactions between C. albicans or N. glabrata H49 and the mouthwash Bexident Post—has previously been documented in the context of antifungal treatments. This phenomenon, often referred to as "paradoxical growth," describes the unexpected ability of certain Candida species to resume growth at high concentrations of antifungal agents, rather than being inhibited. It is a well-recognized mechanism of resistance and is strongly associated with enhanced chitin biosynthesis in the fungal cell wall. Similar responses have also been observed in reaction to certain mouthwashes, suggesting they may trigger comparable resistance mechanisms.”
Comment 14: Results: When referring to "ring-like ghost structures" or "zinc particles," the corresponding figure panels (a, b, c) should be explicitly referenced for clarity.
Response 14 : Done.
Comment 15: Results: While the qualitative SEM analysis is well discussed, it would strengthen the conclusions if quantitative metrics—such as cell wall thickness, pore size, or percentage of disrupted cells—were included, if available.
Response 15: Thank you for your insightful comment. We agree that quantitative metrics such as cell wall thickness, pore size, or percentage of disrupted cells would indeed strengthen the analysis. However, due to the limitations of SEM, such quantitative parameters are challenging to extract with accuracy. These metrics are more readily obtained through techniques like confocal laser scanning microscopy, which was not used in this study. Nevertheless, we have ensured that all relevant and available morphological information observable by SEM has been thoroughly described.
Comment 16: Results: The text inconsistently refers to “Hexitidine”, please change in “Hexetidine” throughout the manuscript.
Response 16 : Done.
Reviewer 3 Report
Comments and Suggestions for Authors
The present study entitled as “Repurposing mouthwashes: antifungal and antibiofilm abilities of commercially available mouthwashes against Candida spp.” focused on investigating antifungal efficacy of commercially available mouthwashes against Candida spp. distributed in the European market. Their results demonstrate that mouthwashes with Chlorhexidine digluconate, either alone or in combination with Cetylpyridinium chloride were the most efficacious formulations than other. Overall, the findings are interesting and show clinical application for the treatment of oral candidiasis associated with the pathogenicity of Candida spp. The manuscript is well-written and well-executed.
However, there are a few minor comments/suggestions which the authors need to address:
- Drug tolerance/resistance becoming very common among various Candida spp. What advantages does this investigation offer in that condition?
- There are studies already published demonstrating the antifungal efficacy of mouthwashes. Discussing how the current study builds upon that work and highlighting its key findings would strengthen the manuscript’s impact and innovation.
- Move the description provided for tables for example, in Table 1 “The strongest inhibitions are marked in green, while the weakest are in red. The white and intermediates tonalities stand for intercalary values.” at the bottom of table under “Note” heading. Same goes for other tables as well.
- Line 129 in result section: Provide full form of MFC for clarity.
- Line 208 in method section: two times “to” repeated.
- The spacing between the words should be consistent throughout the manuscript.
Author Response
Comment 1: Drug tolerance/resistance becoming very common among various Candida spp. What advantages does this investigation offer in that condition?
Response 1 : The central obstacle in combating resistance lies in the rapid adaptability of microorganisms, which outpaces the development of new antifungals. Creating a de-novo antifungal drug demands 15–20 years and $1–3 billion USD, navigating a rigorous 7-stage pipeline from discovery to approval. The ‘repurposable drugs’ approach fits into a world that needs to save resources and find quick solutions for infectious diseases: it is indeed the seek of new therapeutical indications for existing drugs.
Comment 2: There are studies already published demonstrating the antifungal efficacy of mouthwashes. Discussing how the current study builds upon that work and highlighting its key findings would strengthen the manuscript’s impact and innovation.
Response 2 : Unlike most studies in the scientific literature, our research focuses on biofilm testing as a key resistance mechanism, providing insights into how biofilm formation influences antimicrobial efficacy, in the contrary of most of other studies. We add on line 222 a sentence to emphatize this fact.
Comment 3 Move the description provided for tables for example, in Table 1 “The strongest inhibitions are marked in green, while the weakest are in red. The white and intermediates tonalities stand for intercalary values.” at the bottom of table under “Note” heading. Same goes for other tables as well.
Response 3 : Done.
Comment 4: Line 129 in result section: Provide full form of MFC for clarity.
Response 4 : Done
Comment 5: Line 208 in method section: two times “to” repeated.
Response 5 : Done
Comment 6: The spacing between the words should be consistent throughout the manuscript.
Response 6 : Done.
Round 2
Reviewer 2 Report
Comments and Suggestions for Authors
All concerns and queries have been successfully addressed
Author Response
Thank you.